# Domain Generalization Using Large Pre-trained Models With Mixture-of-Adapters

## Abstract

Learning a robust vision model despite large distribution shift situations is an important task for model deployment in real-world settings. Especially, domain generalization (DG) algorithm aims to maintain the performance of a trained model on different distributions which were not seen during training. One of the most effective method has been leveraging the already learned rich knowledge of large pretrained models. However, naively tuning large models to DG tasks is practically infeasible due to memory limitations, extensive time requirements for training, and the risk of learned knowledge deterioration. Parameter-efficient fine-tuning (PEFT) methods have been used to reduce the high computational cost during training and efficiently adapt large models to downstream tasks. In this work for the first time we find that the use of adapters in PEFT methods not only reduce high computational cost during training but also serve as an effective regularizer for DG tasks. Surprisingly, a naive adapter implementation for large models achieve superior performance on common datasets. However, in situations of large distribution shifts, additional factors such as optimal amount of regularization due to the strength of distribution shifts should be considered for a sophisticated adapter implementation. As a result, we propose a mixture-of-expert based adapter fine-tuning method, dubbed as mixture-of-adapters (MoA). We employ multiple adapters that have varying capacities, and by using learnable routers, we allocate each token to a proper adapter. By using both PEFT and MoA methods we effectively alleviate the performance deterioration caused by distribution shifts and achieve state-of-the-art performance on diverse DG benchmarks.

## 1 Introduction

The goal of domain generalization (DG) is to well predict on domains that were unavailable during training (a.k.a unseen domains) (Gulrajani & Lopez-Paz, 2020). In DG settings, the model is trained on multiple source domains and evaluated on one unseen target domain (Zhou et al., 2021; Gulrajani & Lopez-Paz, 2020; Cha et al., 2021). Unlike domain adaptation, domain generalization is unable to access any information about the target domain. Therefore DG algorithms should fully exploit domain invariant features underlying in the source domain to well predict on the target domain (Seo et al., 2020; Gulrajani & Lopez-Paz, 2020; Arjovsky et al., 2019).

In recent times the usage of large pretrained models is gaining popularity in domain generalization fields (Cha et al., 2022; Mao et al., 2022; Lew et al., 2023; Li et al., 2023). Since large pretrained models already possess some extent of domain invariant knowledge (Cha et al., 2022), exploiting this knowledge into domain generalization has become a popular choice. There are few studies that have tried to train these models directly with empirical risk minimization (ERM) (Vapnik, 1998). Angarano et al. (2022) shows that ERM algorithm performs competitively well when accompanied with proper backbones like EfficientNet (Tan & Le, 2019), ViT (Dosovitskiy et al., 2020), DeiT (Touvron et al., 2021), LeViT (Graham et al., 2021) and ConViT (d'Ascoli et al., 2021). This inspired us to leverage large pretrained models on DG settings. However it is widely known that naively fine-tuning a large model is impractical, not only does it demand a large size VRAM caused by high peak memory in training phase, but it also requires a significant amount of training time due to numerous amount of parameters. Additionally, the pretrained feature extractor can be distorted by overfitting on source domains (Gao et al., 2021) and its contextual information can be harmed (Kumar et al., 2022; Mao et al., 2022; Wortsman et al., 2022), which overall results in a generalization

failure to out-of-distribution data. Fig. 1 compares the accuracy of various fine-tuning methods, including naive full fine-tuning, linear probing, and other partial fine-tuning methods,which also highlights their incapability to handle such distribution shifts.

This work represents a pioneering adoption of Parameter-Efficient Fine-Tuning (PEFT) methods in the context of domain generalization, expanding its usage beyond its traditional domains on transfer learning. PEFT methods (Houlsby et al., 2019; Karimi Mahabadi et al., 2021; He et al., 2022; Zaken et al., 2021; Hu et al., 2021; Ryu, 2023; Jia et al., 2022; Chen et al., 2022; Li & Liang, 2021; Sung et al., 2022a) mitigate the high cost of full fine-tuning of large pretrained models and aims to reach or exceed the performance of full fine-tuning or zero-shot performance by tuning only some parts of the model (Zaken et al., 2021) or by employing a small number of external learnable parameters (Hu et al., 2021; Ryu, 2023; Houlsby et al., 2019; Karimi Mahabadi et al., 2021; He et al., 2022; Jia et al., 2022). We compare various aspects of different trainable parameter settings (e.g.,full fine-tuning, and adapter fine-tuning) and empirically verify that PEFT methods can act as an effective regularization during the training process and show that it can largely outperform a fully fine-tuned model and reach comparable performance with recent state-of-the-art methods (Cha et al., 2021; 2022; Arpit et al., 2022; Li et al., 2023) in DG settings. We also discover that the optimal strength of PEFT regularization differs, depending on the amount of a distribution shift.

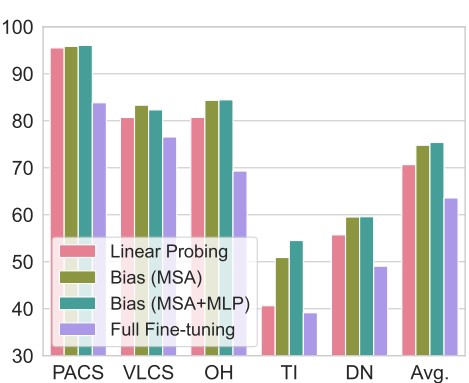

Figure 1: Domain generalization benchmark results with varying trainable parameters in ViT-Base (Dosovitskiy et al., 2020) pretrained with CLIP of OpenAI (Radford et al., 2021). Y axis is accuracy. We use linear probing (denoted as Linear), bias tuning in attention layer (Bias (MSA)), bias tuning in both attention and MLP layer (Bias (MSA+MLP)), and full fine-tuning to show the accuracy change according to the trainable parameter change when using PEFT methods with large models. OH, TI, DN denotes OfficeHome (Venkateswara et al., 2017), TerraIncognita (Beery et al., 2018), and Domain-Net (Peng et al., 2019), respectively.

As shown in Fig. 1, linear probing (Linear), bias tuning in attention layer (Bias (MSA)), bias tuning in both attention and MLP layer (Bias (MSA+MLP)) show comparable performance on PACS (Li et al., 2017) and VLCS Fang et al. (2013) dataset. However on datasets with large distribution shifts, say TerraIncognita (Beery et al., 2018), Bias (MSA+MLP) shows significant improvement compared to the Bias (MSA) and Linear method. This shows that determining the proper amount of regularization by adjusting the trainable parameter is crucial to deal with different distribution shift. To handle this, we introduce mixture-of-expert based adapter architecture, called mixture-of-adapters (MoAs), that adequately manipulate the magnitude of regularization by employing adapters that have different capacity and routing each token to proper adapters. By doing so, we can further improve the performance on DG tasks by employing MoAs and learnable routers to handle the different intensity of distribution shift among various datasets and achieve state-of-the-art performance on DG benchmark.

## 2 RELATED WORK

**Domain generalization.** For the past decade many learning methods on how to learn domain invariant representations have been proposed in the domain generalization field. Empirical risk minimization (ERM) (Vapnik, 1998) which is one of the most simplest approaches, just minimizes the loss on each domains and trains the model. In DomainBed (Gulrajani & Lopez-Paz, 2020), ERM still remains effective within the benchmark's restricted hyperparameter search space and model selection methods. DomainBed tested many DG methods such as IRM (Arjovsky et al., 2019), GroupDRO (Sagawa et al., 2019), MixUp (Xu et al., 2020), DANN (Ganin et al., 2016), and CORAL (Sun & Saenko, 2016) in a unified and contained experimental setup. SWAD (Cha et al., 2021) explored the relationship between flat loss surfaces and DG performance, achieving superior

results. Ensemble-of-Averages (EoA) (Arpit et al., 2022) perform model averaging in training time and ensemble them in test time. Current research is focused on leveraging knowledge from large pretrained models, with MIRO (Cha et al., 2022) introducing an oracle model approximated by a large pretrained model to maximize the mutual information between the target model. Recently, methods that ensemble diverse models show remarkable performance in the DG benchmark. SIMPLE (Li et al., 2023) utilizes many different pretrained models from a ModelPool and extracts outputs from the frozen pretrained models, and trains a shallow dispatcher using these outputs.

**Parameter efficient fine-tuning.** Leveraging large pretrained models for specific tasks involves fine-tuning, but fine-tuning all parameters is impractical. Recent approaches such as Parameter-efficient Fine-Tuning (PEFT) (Houlsby et al., 2019; He et al., 2021; Paul et al., 2022), focus on efficient fine-tuning by freezing most of the model parameters and optimizing only a few for a given task. Many successful PEFT approaches (Zaken et al., 2021; Hu et al., 2021; Ryu, 2023; Gao et al., 2020; He et al., 2021; Jia et al., 2022; Lester et al., 2021; Sung et al., 2022a;b; Zhang et al., 2022; 2023) adopt popular pretrained models to various downstream tasks. Among these methods, the use of adapters (Hu et al., 2021; Karimi Mahabadi et al., 2021; He et al., 2022) are adopted because of its high performance and efficient computation cost. Adapters are small modules trained on specific tasks which are inserted between network layers, where all the layers except for the adapters are frozen during training. LoRA (Hu et al., 2021), Compacter (Karimi Mahabadi et al., 2021) and KAdaptation (He et al., 2022) greatly reduced the number of trainable parameters by attaching a low-rank hypercomplex adapter layers in the transformer model and by decomposing the updated weight matrix into low rank matrices respectively.

**Mixture-of-Experts.** Mixture-of-Experts models (MoEs) are proposed to improve model performance by incorporating multiple subsets of parameters called 'experts' with routing algorithms conditioned by its input (Jacobs et al., 1991; Jordan & Jacobs, 1994; Eigen et al., 2013). Evolving from this, a type of model called sparse MoEs has become popular in both NLP (Shazeer et al., 2017; Lepikhin et al., 2020; Zoph et al., 2022; Fedus et al., 2022; Du et al., 2022) and vision (Ahmed et al., 2016; Gross et al., 2017; Yang et al., 2019; Wang et al., 2020; Riquelme et al., 2021) tasks lately. This is due to its capability to enhance model capacity while simultaneously reducing the substantial increase in the computational resources required for training. In the field of domain generalization, Li et al. (2022) incorporated MoE design into DG tasks, and proposed a Generalizable Mixture-of-Experts (GMoE) architecture to effectively handle distribution shifts.

## 3 PRELIMINARIES

### 3.1 DOMAIN GENERALIZATION

Let us denote the set of training domains as $\mathcal{D} = \{\mathcal{D}_i\}_{i=1}^K$ where $K$ implies the total number of training domains, and $\mathcal{D}_i$ is the distribution over the input space. Also, define the set of target domains as $\mathcal{T} = \{\mathcal{T}_i\}_{i=1}^{K'}$ where $K'$ denotes the total number of training domains. The training dataset is composed of $n_{\mathcal{D}_i}$ data points denoted as $(x_j^i, y_j^i)_{j=1}^{n_{\mathcal{D}_i}} \sim \mathcal{D}_i$ where $x$ is the input and $y$ is the target label for each training domain. In DG settings, the goal is to find the model parameter $\theta$ of a classifier $f_\theta$ which generalizes well on both $\mathcal{D}$ and $\mathcal{T}$. To be more precise, for the ERM algorithm, we define

$$\mathcal{E}_{\mathcal{D}}(\theta) = \frac{1}{K} \sum_{i=1}^K \mathbb{E}_{(x^i, y^i) \sim \mathcal{D}_i}[l(f_\theta(x^i), y^i)] \tag{1}$$

where $l(\cdot, \cdot)$ denote a loss function and define $\mathcal{E}_{\mathcal{T}}$ in the similar manner. We minimize empirical risk $\hat{\mathcal{E}}_{\mathcal{D}}(\theta) = \frac{1}{K} \sum_{i=1}^K \frac{1}{n_{\mathcal{D}_i}} \sum_{j=1}^{n_{\mathcal{D}_i}} [l(f_\theta(x_j^i), y_j^i)]$ during training and expect the optimal parameter $\hat{\theta} = \arg\min_\theta \hat{\mathcal{E}}_{\mathcal{D}}(\theta)$ to be optimal for $\mathcal{E}_{\mathcal{T}}(\theta)$.

### 3.2 PARAMETER EFFICIENT ADAPTER

Parameter-Efficient Fine-Tuning (PEFT) efficiently handles enormous pretrained models to a downstream task and achieves significant performance unlike the direct fine-tuning methods where high cost of memory and computation are required. Previous methods of fine-tuning large models have been accomplished by partial fine-tuning (Touvron et al., 2022) methods. However, recent works

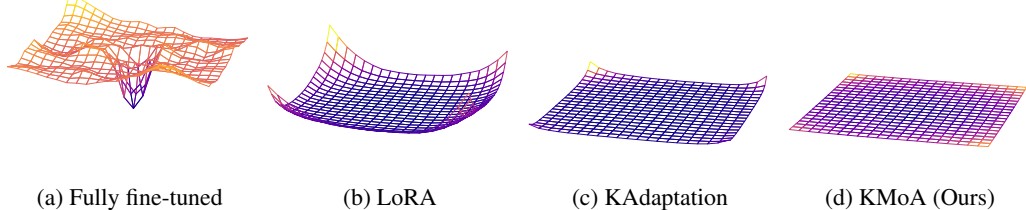

| (a) Fully fine-tuned | (b) LoRA | (c) KAdaptation | (d) KMoA (Ours) |

Figure 2: Flatness comparison of loss surfaces from models trained with full fine-tuning, LoRA, KAdaptation, and KAdaptation with Mixture-of-Adapter (denoted as KMoA) on PACS dataset Li et al. (2017). All visualizations are computed from test environment 0 (Art) domain.

(Houlsby et al., 2019; Hu et al., 2021; Karimi Mahabadi et al., 2021) reveal that employing a learnable layer in the frozen pretrained weights can surpass the performance of the partial fine-tuning method.

More specifically, Aghajanyan et al. (2020) verifies that dense neural network layers with full-rank matrices can be reduced to lower-rank subspaces. From this finding, Hu et al. (2021) constrains the updated weight $\Delta\mathbf{W} = \mathbf{BA} \in \mathbb{R}^{k \times d}$ to have a low intrinsic rank which can be expressed as $\mathbf{h} = \mathbf{W_0}\mathbf{x} + \Delta\mathbf{W}\mathbf{x} = \mathbf{W_0}\mathbf{x} + \mathbf{BAx}$, where $\mathbf{x}$ denotes input sequences of each module, $\mathbf{W_0} \in \mathbb{R}^{k \times d}$ denotes the frozen weight matrix of a pretrained model, and $\mathbf{A} \in \mathbb{R}^{k \times r}$, and $\mathbf{B} \in \mathbb{R}^{r \times d}$ $(r < k, d)$ are the trainable parameters during training.

In addition, KAdaptation (He et al., 2022) and Compacter (Karimi Mahabadi et al., 2021) both exploit Kronecker products to decompose the weight matrices into reduce trainable parameters. The Kroneker product between matrix $\mathbf{A} \in \mathbb{R}^{m \times n}$ and $\mathbf{B} \in \mathbb{R}^{p \times q}$ is denoted by $\mathbf{A} \otimes \mathbf{B} \in \mathbb{R}^{mp \times nq}$ and can be expressed as the following:

$$\mathbf{A} \otimes \mathbf{B} = \begin{pmatrix} a_{11}\mathbf{B} & \dots & a_{1n}\mathbf{B} \\ \vdots & \ddots & \vdots \\ a_{m1}\mathbf{B} & \dots & a_{mm}\mathbf{B} \end{pmatrix} \tag{2}$$

where $a_{ij}$ indicates the element in the $i$-th row and the $j$-th column of $\mathbf{A}$. It decomposes the update weight matrix $\Delta\mathbf{W} = \sum_{i=1}^{t} \mathbf{A}_i \otimes \mathbf{B}_i \in \mathbb{R}^{k \times d}$ where $t$ is a hyperparameter that decides the number of Kronecker products. Furthermore, $\mathbf{W}$ can be expressed as the following: $\mathbf{W} = \sum_{i=1}^{t} \mathbf{A}_i \otimes \mathbf{B}_i = \sum_{i=1}^{n} \mathbf{A}_i \otimes (\mathbf{u}_i\mathbf{v}_i^\top)$ where slow-weights $\mathbf{A}_i \in \mathbb{R}^{t \times t}$ is shared across all layers, and fast-weights $\mathbf{B}_i \in \mathbb{R}^{\frac{k}{t} \times \frac{d}{t}}$ is decomposed into low-rank matrices $\mathbf{u}_i \in \mathbb{R}^{\frac{k}{t} \times r}$ and $\mathbf{v}_i \in \mathbb{R}^{r \times \frac{d}{t}}$ with $i \in \{1, \dots, t\}$. Compacter decomposes the weight matrix to the additional adapter layer whereas KAdaptation decomposes the update matrix $\Delta\mathbf{W}$ in the original layer.

## 4 ANALYSIS OF ADAPTERS FOR DOMAIN GENERALIZATION

In this section, we investigate how adapters affect loss landscape flatness by analyzing the maximum Hessian eigenvalue spectra and loss landscapes of various trained models. It is widely known that finding a flat minima during optimization is closely related to the generalization performance and robustness of trained models, as documented in prior studies (Izmailov et al., 2018; Keskar et al., 2016; Garipov et al., 2018; Foret et al., 2020; Cha et al., 2021; Park & Kim, 2022). Therefore, by comparing loss landscapes along with maximum Hessian eigenvalues, we can anticipate which model will generalize better to unseen domains. We show the results of our analysis in the following sections.

### 4.1 LOSS LANDSCAPES

As shown in (Park & Kim, 2022; Garipov et al., 2018; Li et al., 2018) randomly perturbing the trained weights with random direction vectors, we can obtain variations of the loss value, which can be used to predict the models generalization ability. We evaluate on PACS (Li et al., 2017) dataset, one of the dataset in the DG benchmark (Gulrajani & Lopez-Paz, 2020). As shown in Fig. 2,

Table 1: Quantitative evaluation on domain generalization datasets with different PEFT methods. Following DomainBed (Gulrajani & Lopez-Paz, 2020), we evaluate our algorithm on PACS (Li et al., 2017), VLCS (Fang et al., 2013), OfficeHome (Venkateswara et al., 2017), TerraIncognita (Beery et al., 2018) and DomainNet (Peng et al., 2019). For fair comparison, we use training-domain-validation model selection strategy. Reported performance of each algorithms are from their papers. We indicate the pretraining method and dataset as the form of PRETRAIN$_{\text{DATASET}}$, and IG3B denotes Instagram-3B dataset from Zellers et al. (2018). Results with our method are colored in blue.

| Algorithm | Architecture | Pretraining | PACS | VLCS | OfficeHome | TerraIn. | DomainNet | Avg. | #Param. | Trainable #Param. |
|---|---|---|---|---|---|---|---|---|---|---|
| MIRO | RegNetY-16GF | SWAG$_{\text{IG3B}}$ | 97.4 $\pm$ 0.2 | 79.9 $\pm$ 0.6 | 80.4 $\pm$ 0.2 | 58.9 $\pm$ 1.3 | 53.8 $\pm$ 0.1 | 74.1 | 167.2M | 83.6M |
| SMA | RegNetY-16GF | SWAG$_{\text{IG3B}}$ | 95.5 $\pm$ 0.0 | 80.7 $\pm$ 0.1 | 82.0 $\pm$ 0.0 | 59.7 $\pm$ 0.0 | 60.0 $\pm$ 0.0 | 75.6 | 83.6M | 83.6M |
| MIRO+SWAD | RegNetY-16GF | SWAG$_{\text{IG3B}}$ | 96.8 $\pm$ 0.2 | 81.7 $\pm$ 0.1 | 83.3 $\pm$ 0.1 | **64.3** $\pm$ 0.3 | 60.7 $\pm$ 0.0 | **77.3** | 167.2M | 83.6M |
| *Methods with Parameter-Efficient Fine-Tuning (PEFT)* | | | | | | | | | | |
| ERM (Baseline) | ViT-B/16 | CLIP$_{\text{LAION2B}}$ | 85.8 $\pm$ 2.1 | 78.5 $\pm$ 0.9 | 78.1 $\pm$ 0.8 | 41.0 $\pm$ 1.6 | 52.2 $\pm$0.1 | 67.1 | 85.8M | 85.8M |
| ERM$_{\text{LoRA},r=2}$ | ViT-B/16 | CLIP$_{\text{LAION2B}}$ | 96.4 $\pm$ 0.6 | 82.6 $\pm$ 0.6 | 86.7 $\pm$ 0.3 | 46.1 $\pm$1.7 | 61.5 $\pm$0.1 | 74.7 | 85.9M | 0.11M |
| ERM$_{\text{KAdaptaion}}$ | ViT-B/16 | CLIP$_{\text{LAION2B}}$ | **97.5** $\pm$ 0.1 | **83.0** $\pm$ 0.1 | **90.3** $\pm$ 0.1 | 51.9 $\pm$0.5 | **62.7** $\pm$0.0 | 77.1 | 85.9M | 0.14M |

compared to the case of full fine-tuning (Fig. 2a) where we trained all of the model parameters, model parameters tuned with LoRA (Hu et al., 2021; Ryu, 2023) (Fig. 2b) and KAdaptation (He et al., 2022) (Fig. 2c) displays a more flatter loss surface. Additionally, KAdaptation shows a flatter loss surface than LoRA. We show further visualizations of the loss landscape on additional test environments on PACS in Appendix A.1.

## 4.2 MAXIMUM HESSIAN EIGENVALUE SPECTRA

Drawing loss landscapes from the optimal point to random directions sometimes fail to fully represent the shape of loss surfaces due to the high dimensionality of the loss surface (Garipov et al., 2018; Li et al., 2018). Therefore, following Park & Kim (2022) we calculate the top-5 Hessian eigenvalues and show their spectra.

Analogous to loss landscape results in the previous section, we observe the same phenomena with the use of adapters, say LoRA and KAdaptation. As depicted in Fig. 3, the top-5 Hessian eigenvalues are further zero-concentrated than the fully fine-tuned model. These discrepancy implies that employing adapters with large pretrained models results in a flatter loss surface around the optimal point. This is because the Hessian matrix serves as an indicator of the curvature of the loss surface. Since the flatness of loss surfaces has a substantial impact on model performance in DG tasks Cha et al. (2021), it can be concluded that adapter fine-tuning with large pretrained models effectively facilitate the performance in domain generalization.

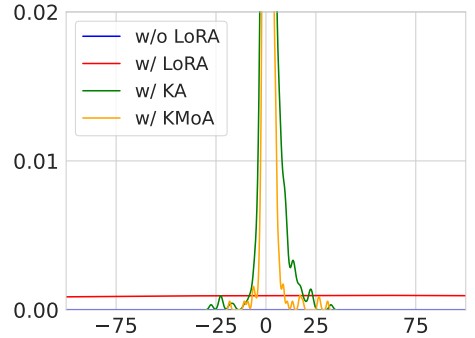

Figure 3: Hessian max eigenvalue spectra of test environment 0 (Art) in PACS dataset. KA and KMoA denotes KAdaptation and Mixture-of-Adapter with KAdaptation method, respectively. In the order of w/o LoRA, w/LoRA, w/KA, and w/KMoA, the hessian eigenvalue spectrum concentrates to zero and eigenvalues are suppressed accordingly.

We show visualizations of max Hessian eigenvalue spectra on additional test environments on PACS dataset in Appendix A.2.

## 4.3 PARAMETER-EFFICIENT ADAPTER FOR DOMAIN GENERALIZATION

Previously, we demonstrated that models trained with parameter-efficient fine-tuning methods have the potential to reach a more generalizable optimization point. Nevertheless, it's challenging to come to a conclusive decision that a flatter loss surface directly translates to better generalization performance. Consequently, in the following section, we conduct a comprehensive evaluation of various fine-tuning methods and make comparisons between them. Finally, we discuss about the most effective practical fine-tuning approach and explore strategies to further enhance performance.

In Tab. 1 we show results from various paper on five datasets, namely PACS (Li et al., 2017), VLCS (Fang et al., 2013), OfficeHome (Venkateswara et al., 2017), TerraIncognita Beery et al. (2018) and DomainNet (Peng et al., 2019), and from the observations above, we evaluate ERM with various PEFT adapter methods in standard DG benchmark form Gulrajani & Lopez-Paz (2020). Some works like Cha et al. (2022) report the results of full fine-tuning a ViT-B/16 model with an ERM algorithm. However just by training a small adapter layer or an attention layer, we observe a greater performance improvement. Compared to the methods that use additional regularization or ensembling, our parameter efficient training approach with naive ERM algorithm reaches comparable accuracy and even state-of-the-art for the OfficeHome dataset. Additionally KAdaptation method achieves the best average accuracy of all the methods. These results validate our observation in Sec. 4.1 and 4.2, demonstrating that a proper selection of an adapter method can boost the performance adequately.

## 5 MIXTURE-OF-ADAPTERS (MOA)

In this section we show the effectiveness of our proposed method, mixture-of-adapter, which can tune large models with low cost and achieve great performance on DG tasks. As revealed in Li et al. (2022), mixture-of-expert model architectures with the use of cosine routing is more effective on DG tasks. Experts also handle some parts of the visual attributes, therefore reducing incorrect token allocation caused by intra-domain similarities. We bring it to our adapter-based method, which maintains the computational efficiency and achieves better results in domain generalization.

As demonstrated in Sec. 1, we utilize adapters to handle various amounts of distribution shift among the domains. Specifically, adapters, each with distinct capacities at every MoA layer, are employed to select the top-k outputs which are averaged and then integrated with the original layer's outputs. We manipulate the capacities of each adapter by adjusting its inner rank.

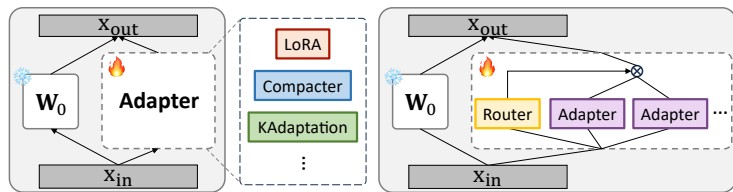

(a) Adapter PEFT methods     (b) Mixture-of-Adapter method

Figure 4: Architecture of proposed Mixture-of-Adapters (MoA). $\mathbf{W}_0$, $\mathbf{x}_{\text{in}}$, and $\mathbf{x}_{\text{out}}$ denotes original pretrained weight, input, and output tokens in multi-head self-attention (MHSA). `Adapter` in Fig. 4a can be any adapter-based PEFT methods like LoRA, Compacter, KAdatptaion, etc, and `Router` in Fig. 4b can be linear or cosine router that commonly used in Mixture-of-Expert methods.

The mixture-of-experts (MoE) layer utilizing the cosine router $\text{G}(\mathbf{x})$ with embedding $\mathbf{E}$ and adapter $\text{A}_{r_i}$ with inner rank $r_i$ can be denoted as:

$$
\begin{aligned}
f_{\text{MoE}}(\mathbf{x}) &= \sum_{i=1}^{N} \text{G}(\mathbf{x})_i \text{A}_{r_i}(\mathbf{x}) \\
&= \sum_{i=1}^{N} \text{TOP}_k \left( \text{Softmax} \left( \frac{\mathbf{E}^{\text{T}}\mathbf{W}\mathbf{x}}{\tau \|\mathbf{W}\mathbf{x}\|\|\mathbf{E}\|} \right) \right)_i \text{A}_{r_i}(\mathbf{x})
\end{aligned}
\tag{3}
$$

where $\text{A}_{r_i}(\mathbf{x})$ is the output of an adapter with a different inner rank $r_i$ and $\tau$ is a learnable temperature term of a softmax layer. In detail, we incorporate adapters in the large pretrained model by attaching them to attention sub modules. Following the implementation of traditional mixture-of-expert (MoE) methods, routers are adopted to dispatch tokens to the appropriate expert adapters. An adapter can be of any form such as LoRA (Hu et al., 2021), one of the most popular method in PEFT, Compacter (Karimi Mahabadi et al., 2021), and KAdaptation (He et al., 2022). In the case of KAdaptation, we choose the inner rank $r_i$ as the additional decomposition of $\mathbf{B}_{\text{i}}$. Visualizations of our architecture is shown in Fig. 4

Table 2: Quantitative evaluation on domain generalization datasets with Mixture-of-Adapter (MoA) methods. The numbers marked with $^\dagger$ are reported results from Cha et al. (2022). OpenAI denotes the private dataset used to train CLIP (Radford et al., 2021), and ModelPool-A and B denotes the set of pretrained models used in Li et al. (2023). Results with ensemble methods are colored in gray, and results with our method are colored in blue.

| Algorithm | Architecture | Pretraining | PACS | VLCS | OfficeHome | TerraIn. | DomainNet | Avg. | #Param. | Trainable #Param. |
|---|---|---|---|---|---|---|---|---|---|---|
| ERM$^\dagger$ | ViT-B/16 | CLIP$_{OpenAI}$ | 83.4 $\pm$ 0.5 | 75.9 $\pm$ 1.3 | 66.4 $\pm$ 0.5 | 35.3 $\pm$ 0.8 | 44.4 $\pm$ 0.6 | 61.1 | 85.8M | 85.8M |
| MIRO$^\dagger$ | ViT-B/16 | CLIP$_{OpenAI}$ | 95.6 $\pm$ 0.8 | 82.2 $\pm$ 0.3 | 82.5 $\pm$ 0.1 | 54.3 $\pm$ 0.4 | 54.0 $\pm$ 0.3 | 73.7 | 172M | 85.8M |
| ERM$^\dagger$ | RegNetY-16GF | SWAG$_{IG3B}$ | 89.6 $\pm$ 0.4 | 78.6 $\pm$ 0.3 | 71.9 $\pm$ 0.6 | 51.4 $\pm$ 1.8 | 48.5 $\pm$ 0.6 | 68.0 | 83.6M | 83.6M |
| MIRO$^\dagger$ | RegNetY-16GF | SWAG$_{IG3B}$ | 97.4 $\pm$ 0.2 | 79.9 $\pm$ 0.6 | 80.4 $\pm$ 0.2 | 58.9 $\pm$ 1.3 | 53.8 $\pm$ 0.1 | 74.1 | 167.2M | 83.6M |
| ERM+SWAD$^\dagger$ | RegNetY-16GF | SWAG$_{IG3B}$ | 94.7 $\pm$ 0.2 | 79.7 $\pm$ 0.2 | 80.0 $\pm$ 0.1 | 57.9 $\pm$ 0.7 | 53.6 $\pm$ 0.6 | 73.2 | 83.6M | 83.6M |
| MIRO+SWAD$^\dagger$ | RegNetY-16GF | SWAG$_{IG3B}$ | 96.8 $\pm$ 0.2 | 81.7 $\pm$ 0.1 | 83.3 $\pm$ 0.1 | 64.3 $\pm$ 0.3 | 60.7 $\pm$ 0.0 | 77.3 | 167.2M | 83.6M |
| SMA | RegNetY-16GF | SWAG$_{IG3B}$ | 95.5 $\pm$ 0.0 | 80.7 $\pm$ 0.1 | 82.0 $\pm$ 0.0 | 59.7 $\pm$ 0.0 | 60.0 $\pm$ 0.0 | 75.6 | 83.6M | 83.6M |
| *Methods with Parameter-Efficient Fine-Tuning (PEFT)* | | | | | | | | | | |
| ERM (Baseline) | ViT-B/16 | CLIP$_{LAION2B}$ | 85.8 $\pm$ 2.1 | 78.5 $\pm$ 0.9 | 78.1 $\pm$ 0.8 | 41.0 $\pm$ 1.6 | 52.2 $\pm$ 0.1 | 67.1 | 85.8M | 85.8M |
| ERM$_{Compacter}$ | ViT-B/16 | CLIP$_{LAION2B}$ | 94.1 $\pm$ 0.4 | 81.0 $\pm$ 0.5 | 83.0 $\pm$ 0.1 | 35.9 $\pm$ 0.7 | 56.2 $\pm$ 1.2 | 70.0 | 85.9M | 0.10M |
| ERM$_{Attention}$ | ViT-B/16 | CLIP$_{LAION2B}$ | 93.8 $\pm$ 0.6 | 82.0 $\pm$ 0.3 | 85.9 $\pm$ 0.4 | 51.4 $\pm$ 0.8 | 57.2 $\pm$ 0.1 | 74.1 | 85.8M | 28.4M |
| ERM$_{LoRA,r=2}$ | ViT-B/16 | CLIP$_{LAION2B}$ | 96.4 $\pm$ 0.6 | 82.6 $\pm$ 0.6 | 86.7 $\pm$ 0.3 | 46.1 $\pm$ 1.7 | 61.5 $\pm$ 0.1 | 74.7 | 85.9M | 0.11M |
| ERM$_{KAdaptaion}$ | ViT-B/16 | CLIP$_{LAION2B}$ | 97.5 $\pm$ 0.1 | 83.0 $\pm$ 0.1 | 90.3 $\pm$ 0.1 | 51.9 $\pm$ 0.5 | 62.7 $\pm$ 0.0 | 77.1 | 85.9M | 0.14M |
| *Methods with Mixture-of-Adapter* | | | | | | | | | | |
| ERM$_{LoRA-MoA}$ | ViT-B/16 | CLIP$_{LAION2B}$ | 96.9 $\pm$ 0.3 | 82.8 $\pm$ 0.7 | 89.5 $\pm$ 0.2 | 49.2 $\pm$ 2.4 | 62.2 $\pm$ 0.0 | 75.9 | 87.2M | 1.5M |
| ERM$_{KAdaptaion-MoA}$ | ViT-B/16 | CLIP$_{LAION2B}$ | 97.4 $\pm$ 0.2 | 83.1 $\pm$ 0.3 | 90.6 $\pm$ 0.0 | 52.8 $\pm$ 1.4 | 62.7 $\pm$ 0.1 | 77.3 | 87.3M | 1.5M |
| *Methods Using Ensemble* | | | | | | | | | | |
| EoA | RegNetY-16GF | SWAG$_{IG3B}$ | 95.8 | 81.1 | 83.9 | **61.1** | 60.9 | 76.6 | > 500M | - |
| SIMPLE | ModelPool-A | ModelPool-A | 88.6 $\pm$ 0.4 | 79.9 $\pm$ 0.5 | 84.6 $\pm$ 0.5 | 57.6 $\pm$ 0.8 | 49.2 $\pm$ 1.1 | 72.0 | > 1,000M | 0.9M |
| SIMPLE$^+$ | ModelPool-B | ModelPool-B | **99.0** $\pm$ 0.1 | 82.7 $\pm$ 0.4 | 87.7 $\pm$ 0.4 | 59.0 $\pm$ 0.6 | 61.9 $\pm$ 0.5 | 78.1 | > 1,000M | 0.9M |
| ERM$_{KAdaptaion-MoA-Ensemble}$ | ViT-B/16 | CLIP$_{LAION2B}$ | 97.6 | **83.4** | **90.9** | 54.3 | **63.1** | 77.9 | 261.3M | - |

# 6 EXPERIMENTS

**Experimental details.** We use the standard benchmark DomainBed (Gulrajani & Lopez-Paz, 2020) for training and evaluating the performance on the domain generalization task. Following this, we use fixed hyperparameters within the same backbone model for all the experiments. We train five types of models which are full fine-tuning, attention tuning, LoRA (Hu et al., 2021), KAdaptation (He et al., 2022), and Compacter (Karimi Mahabadi et al., 2021). We employ CLIP (Radford et al., 2021) trained ViT (Dosovitskiy et al., 2020) as our initialization model because CLIP carries strong zero-shot ability which can be used to obtain favorable generalization performance. In detail we use OpenCLIP (Ilharco et al., 2021), an open-source re-implementation of CLIP model with LAION-2B dataset (Schuhmann et al., 2022), for all our experiments. We describe additional implementation details in Appendix B.1

**Evaluation protocols and datasets.** For a fair comparison, we employ DomainBed evaluation protocols (Cha et al., 2021; Gulrajani & Lopez-Paz, 2020). The following five benchmark datasets are used: PACS (Li et al., 2017), VLCS (Fang et al., 2013), OfficeHome (Venkateswara et al., 2017), TerraIncognita (Beery et al., 2018), and DomainNet (Peng et al., 2019). Using a *leave-one-out cross-validation*, all performance scores are evaluated by averaging all the cases that use a single domain as the target domain and the others as the source domains. Experiment is repeated three times and 20% percent of source domain data is left out for validation purposes. Lastly model selection (training-domain validation) and hyperparameter search procedure is referenced from DomainBed (Gulrajani & Lopez-Paz, 2020). We perform three runs with different random seeds for each setting and report their mean and standard deviation to show the training randomness. In ablation studies, we keep all the random seeds fixed and conduct the experiment.

## 6.1 RESULTS FOR MIXTURE-OF-ADAPTER

As described in Sec. 5, we implement Mixture-of-Adapter (MoA) with LoRA (Hu et al., 2021) and KAdaptation, and report the benchmark result in Table 2. In addition to the performance gains by applying PEFT, results after employing MoA are increased consistently at all datasets. In the case of LoRA-MoA, average performance is increased by 1.2pp compared to the results when using only LoRA. Also, KAdaptation-MoA method achieves state-of-the-art results on VLCS, OfficeHome, DomainNet datasets, exceeding the result of the original KAdaptation by 0.2pp in average, validating the analyses about loss surface in Fig. 2. Additionally, following Arpit et al. (2022), ensembling the three weights obtained from different seeds, we further enhance the performace by 0.6pp in average, achieving best results in VLCS, OfficeHome, and DomainNet.

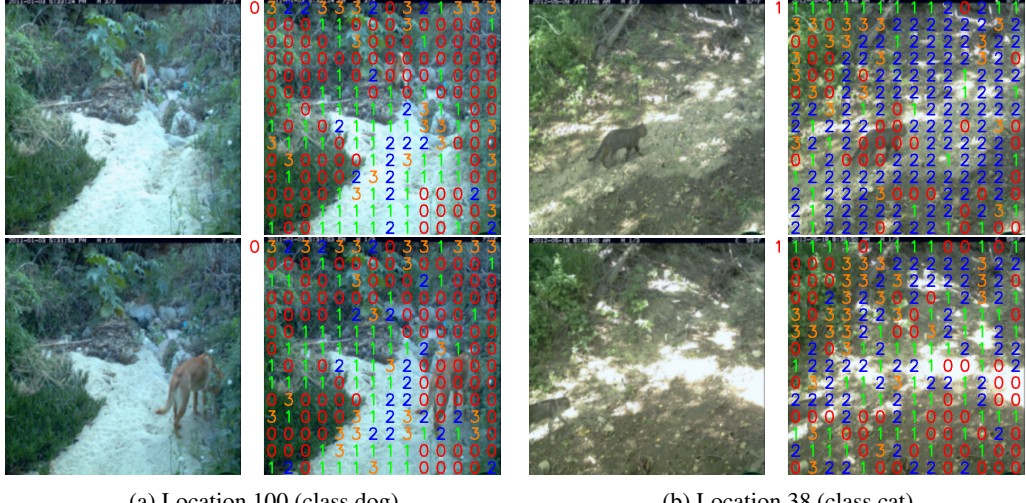

(a) Location 100 (class dog)         (b) Location 38 (class cat)

Figure 5: Visualizations of routed indices of each patch in TerraIncognita (Beery et al., 2018) dataset. Left column is original image, and in right column we indicate where each patch is routed. Upper and lower images were took from same location but different time, therefore they have same background but different object (dog, cat) shape and location.

**Analysis on the router and experts**   To understand why using multiple adapters with a router improves performance, we conduct experiments on what the router see and what each experts learn. To investigate the role of an adapter, we monitor the routing path of each token and identify the expert to which it is directed. This visualization is performed on TerraIncognita dataset. In Fig. 5, the image is divided into patches and each patch is assigned a number, where the number corresponds to the routed expert. We also reveal that the router tends to cluster tokens in areas with semantic information (e.g., object foreground or object outlines). As an example, consider the dog depicted in Fig. 5a and the cat in Fig. 5b, which exhibit varying positions between the first and second rows. Images sharing the same location have consistent backgrounds, resulting in shared expert routing for background tokens. However, objects and their positions can differ, causing object-related tokens to be routed to different experts. In conclusion, the importance of routing numerous tokens to their respective adapters greatly enhances the model's ability to capture semantic information in images, thereby enabling the model to effectively navigate challenging distribution shift scenarios in domain generalization tasks. We show more visualizations of routed patches in Appendix A.3.

## 6.2 ABLATION STUDY

**Effect of fine-tuning in pretrained models**   We observe that using a fine-tuned model with a smaller dataset shows degraded performance. Specifically, LAION-2B (Schuhmann et al., 2022) pretrained CLIP-ViT model almost consistently outperforms the LAION-2B pretrained, ImageNet Deng et al. (2009) fine-tuned CLIP-ViT model across all the adapter methods in terms of accuracy except for TerraIncognita dataset, as illustrated in Fig. 6. This can be attributed to the superior generalization capabilities of larger models and the adapter's capacity to preserve the knowledge from the pretrained model. These findings align with previous literature, such as Kumar et al. (2022), which suggests that fine-tuning entire large models can degrade the learned representations.

**Ablation study on routing strategy in Mixture-of-Adapter**   We conduct an ablation study on three components proposed in Li et al. (2022), which are cosine router, auxiliary loss $\mathcal{L}_{\text{aux}}$, and the layer configuration on the location of the MoE layer. Li et al. (2022) demonstrates that employing a cosine router in their Generalizable Mixture-of-Experts (GMoE) architecture yields better performance when contrasted with employing a linear router. Our approach significantly deviates from theirs, as each of our expert adapters exhibit distinct capacities, and our MoA is attached to an attention layer. Thus we test both the linear and cosine routers. The auxiliary loss function ($\mathcal{L}_{\text{aux}}$) balances the amount of token allocation to each experts, and the layer configuration called 'Every 2' or 'Last 2' in their work denotes the attachment of MoE layer with every two layers or only on

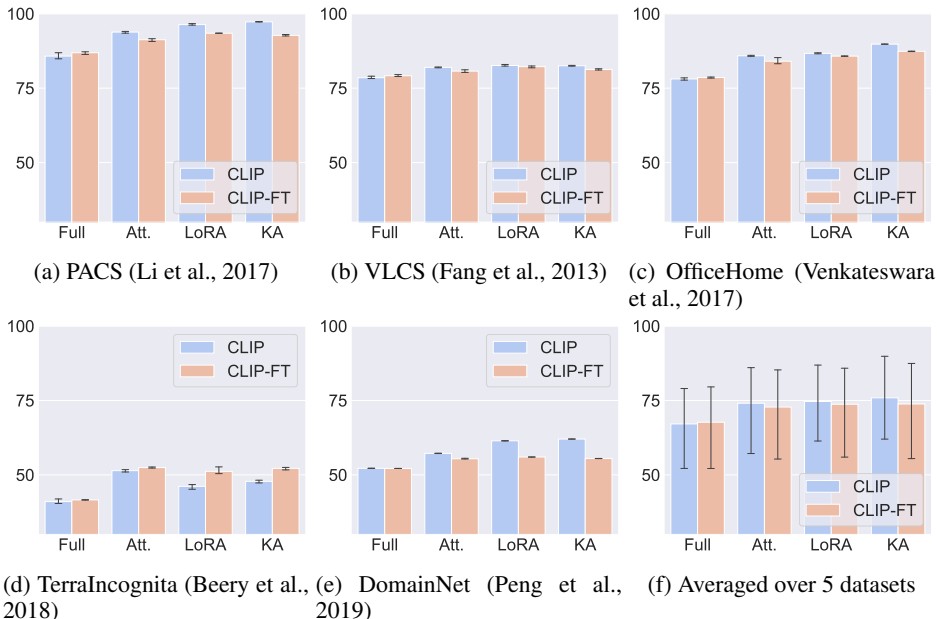

Figure 6: Performance comparison between original (not fine-tuned) CLIP and ImageNet fine-tuned CLIP from timm (Wightman, 2019) (denoted as CLIP-FT) on different fine-tuning strategies. Full, Att., LoRA, and KA denotes full fine-tuning, attention-only tuning (Touvron et al., 2022), LoRA, and KAdaptation, respectively.

Table 3: Performance comparison on different mixture-of-adapter settings. We perform experiments with dropping each components independently from our best setting which is highlighted in blue. 'w/o $\mathcal{L}_{\text{aux}}$' denotes the results without auxiliary loss that used in (Li et al., 2022), 'Cosine→Linear' denotes the result when we changed the Cosine router to a Linear router, and 'Every→Last' denotes the result when we changed the adapter attached layer from Every, which attaches adapters in every two layer, to Last, which attaches adapters in last two layers.

| Adapter | Changed component | PACS | VLCS | OfficeHome | TerraIn. | DomainNet | Avg. | #Param. | Trainable #Param. |
|---|---|---|---|---|---|---|---|---|---|
| | Original | **97.5** | **82.8** | **90.6** | 53.1 | **62.6** | 77.3 | 87.3M | 1.5M |
| KAdaptaion-MoA | w/o $\mathcal{L}_{\text{aux}}$ | 97.3 | 82.6 | 90.5 | **54.0** | 62.6 | 77.4 | 87.3M | 1.5M |
| | Cosine → Linear | 97.5 | 82.3 | 90.3 | 51.5 | 62.6 | 76.9 | 86.1M | 0.33M |
| | Every → Last | 97.2 | 82.2 | 90.2 | 47.5 | 62.4 | 75.9 | 86.3M | 0.51M |

the last two layers. Therefore we also conduct experiments on these settings and report the whole results in Table 3. In our setting, employing a cosine router showed a consistent performance increase for all domains. Also attaching an adapter with every two layers brought better performance than using MoA only on the last two layers. While there was a minimal difference between using and not using $\mathcal{L}_{\text{aux}}$ (within the error margin), using the auxiliary loss yielded more better results for the four datasets (PACS, VLCS, OfficeHome, DomainNet). Thus we decided to incorporate the auxiliary loss $\mathcal{L}_{\text{aux}}$ when conducting our main experiment.

## 7 CONCLUSION

We have shown that using only parameter-efficient fine-tuning can outperform or be competitive with previous state-of-the-art domain generalization algorithms. Additionally, we propose integrating extra adapters with learnable routers to handle various distribution shift situations. This allows us to achieve state-of-the-art results without the need for ensemble methods, and when we do employ ensembling, our results are on par with those achieved by ensemble approaches. We observe the powerful efficacy of PEFT and large models about domain generalization during this work, and expect that this research can motivate the way of research directions of domain generalization towards robustly fine-tuning the large pretrained models.

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

# Appendix

In this appendix, we provide a more detailed explanations of the implementation details, results and visualizations of additional analyses, and discuss limitations of this work.

## A  ADDITIONAL ANALYSIS

In this section, we show additional analysis about loss landscapes, max Hessian eigenvalue spectra, and analyses about routed tokens.

### A.1  COMPARISONS OF LOSS LANDSCAPE VISUALIZATIONS

We show loss landscapes for all test environments in PACS dataset (Li et al., 2017) in Fig. 7. Similar with the visualizations in main paper, the other test environments have a tendency that fully fine-tuned models show most sharp loss landscape. But trained models with LoRA and KAdaptation shows much more flatter loss landscapes, especially KAdaptation have most flat loss landscape.

### A.2  COMPARISONS OF MAX HESSIAN EIGENVALUE SPECTRA

We show the max Hessian eigenvalue spectra evaluated on the all domains from PACS dataset (Li et al., 2017) in Fig. 8. Every domain shows consistent shape of max Hessian eigenvalue spectra that they shows more zero-concentrated shape in the order of w/o LoRA, w/LoRA, w/ KA and w/KMoA.

### A.3  VISUALIZATIONS OF ROUTED PATCHES IN PACS AND TERRAINCOGNITA DATASET.

We additionally show the visualizations of routed patch indices in Fig. 9 on PACS dataset (Li et al., 2017), and Fig. 10, 11 on TerraIncognita dataset (Beery et al., 2018). All images are visualizations from the last adapter-attached transformer layer, layer 10. Similar with the findings from main paper, we can observe that same indices are clustered at the regions where having semantic meanings.

## B  ADDITIONAL EXPERIMENTAL DETAILS

### B.1  IMPLEMENTATION DETAILS

Pretrained vision transformer models are obtained from the timm (Wightman, 2019) library. Adam (Kingma & Ba, 2014) optimizer is used for model optimization along with a learning rate of $5e - 5$. A batch size of 32 per domain is used for the ViT-Base model. We run 15,000 iterations on DomainNet and 5,000 for others, and evaluate at every 500 iteration steps for DomainNet, 200 steps for others. For the adapters, we arbitrarily choose the inner rank so that all adapter methods have approximately the same number of trainable parameters ($\sim$0.1M). The LoRA (Hu et al., 2021) and KAdaptation (He et al., 2022) weights are attached to the QKV projection matrix and projection weight matrix in the self-attention layer. Also for Compacter (Karimi Mahabadi et al., 2021), we attach the adapter after the MLP layer in every blocks. Every adapter except LoRA are implemented using their official repository, and for LoRA, we implement it using unofficial implementation (Ryu, 2023). For pretrained models, we use models from timm (Wightman, 2019) library for all our experiments, and we perform all experiment on one server with 8 NVIDIA RTX3090 GPUs. We'll release training code for all our experiments and trained weights in public.

### B.2  DETAILS ABOUT HYPERPARAMETERS

We report all the hyperparameters that are used in our experiments and analysis in Table 4.

## C  LIMITATIONS

Our method heavily relies on the performance of large pretrained models, hence using a better pretrained model can lead to improved performance. But, such models are limited and require a sub-

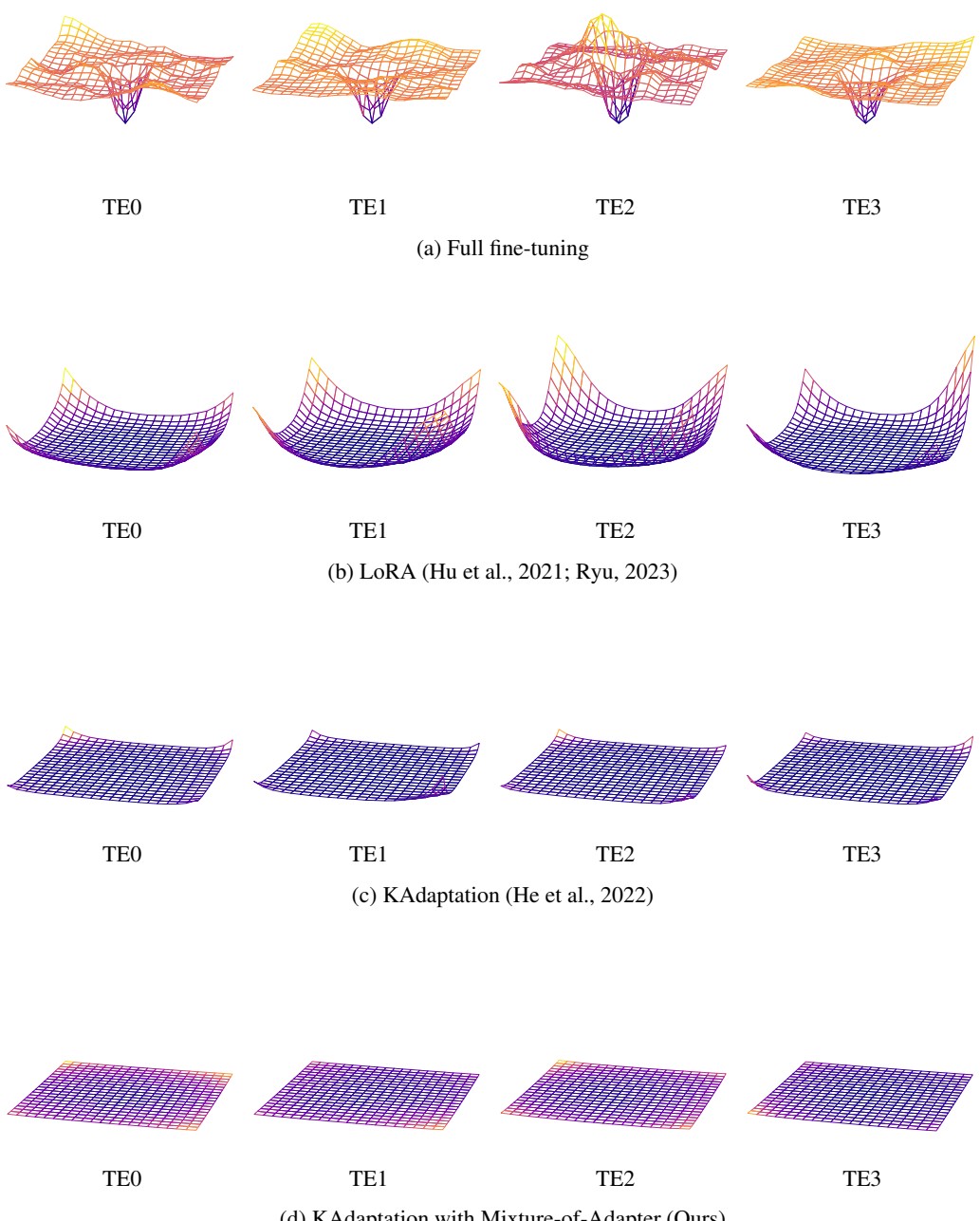

Figure 7: Flatness comparison of loss surfaces trained with full fine-tuning, LoRA, KAdaptation, and KAdaptation with mixture-of-expert on PACS dataset (Li et al., 2017).

stantial amount of time and cost for training. These weakness also exist in methods like MIRO (Cha et al., 2022) or SIMPLE (Li et al., 2023), and the availability of high-performance open-source models like OpenCLIP (Ilharco et al., 2021) can alleviate these drawbacks. Our approach may not yield significantly better performance on datasets that are more challenging than TerraIncognita (Beery et al., 2018) since it has fewer trainable parameters compared to fully fine-tuned DG algorithms. However, our method offers flexibility in adjusting trainable parameters through the control of inner

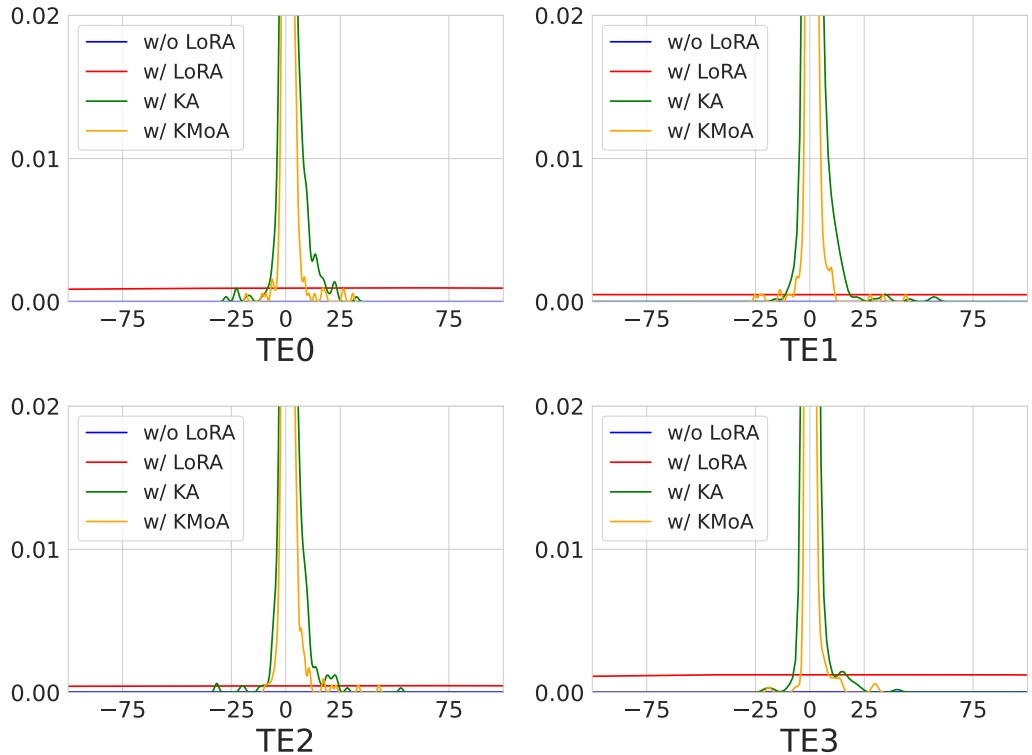

Figure 8: Comparison of max Hessian eigenvalue spectra trained with full fine-tuning, LoRA, and KAdaptation on PACS dataset (Li et al., 2017). In all test environments we can observe that our proposed KMoA shows the most zero-concentrated eigenvalue specturm, and other two adapter method, KA and LoRA also shows smaller max Hessian eigenvalue distribution compared to the fully fine-tuned case (w/o LoRA).

Table 4: List of hyperparameters used in experiments on domain generalization benchmarks.

| Hyperparameter | LoRA | KAdaptation | LoRA-MoA | KMoA |
|---|---|---|---|---|
| # of Experts | | N/A | 4 | 4 |
| Router | | N/A | Cosine | |
| Router Top-k | | N/A | Top-1 | |
| Rank of adapter ($r_i$) | 2 | 2 | $[1, 2, 4, 8]$ | |
| # of Kroneker products ($t$) | N/A | 64 | N/A | 64 |
| Batch size | | 160 (DomainNet), 96 (Otherwise) | | |
| Learning rate | | $5e-5$ | | |
| Optimizer | | Adam | | |

rank $r_i$ and allows for obtaining the optimal rank through hyperparameter search, addressing this limitation effectively.

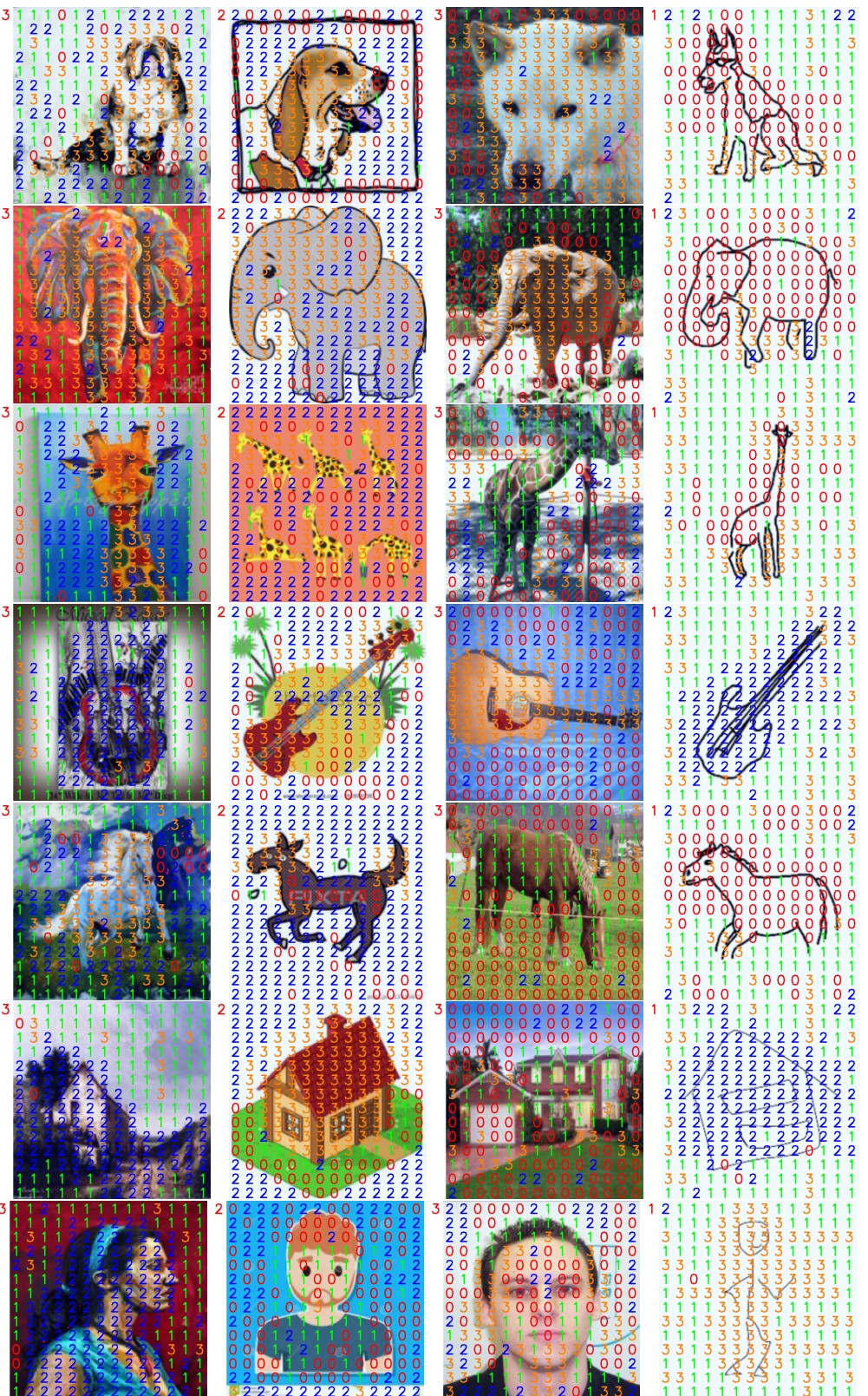

Figure 9: Visualizations of routed indices of each patch. We show a total of seven classes in PACS dataset (Li et al., 2017), with one class per row in the order of 'Dog', 'Elephant', 'Giraffe', 'Guitar', 'Horse', 'House', 'Person'. Also, in each column, the same domains are located in the order of 'Art Painting', 'Cartoon', 'Photo', and 'Sketch'.

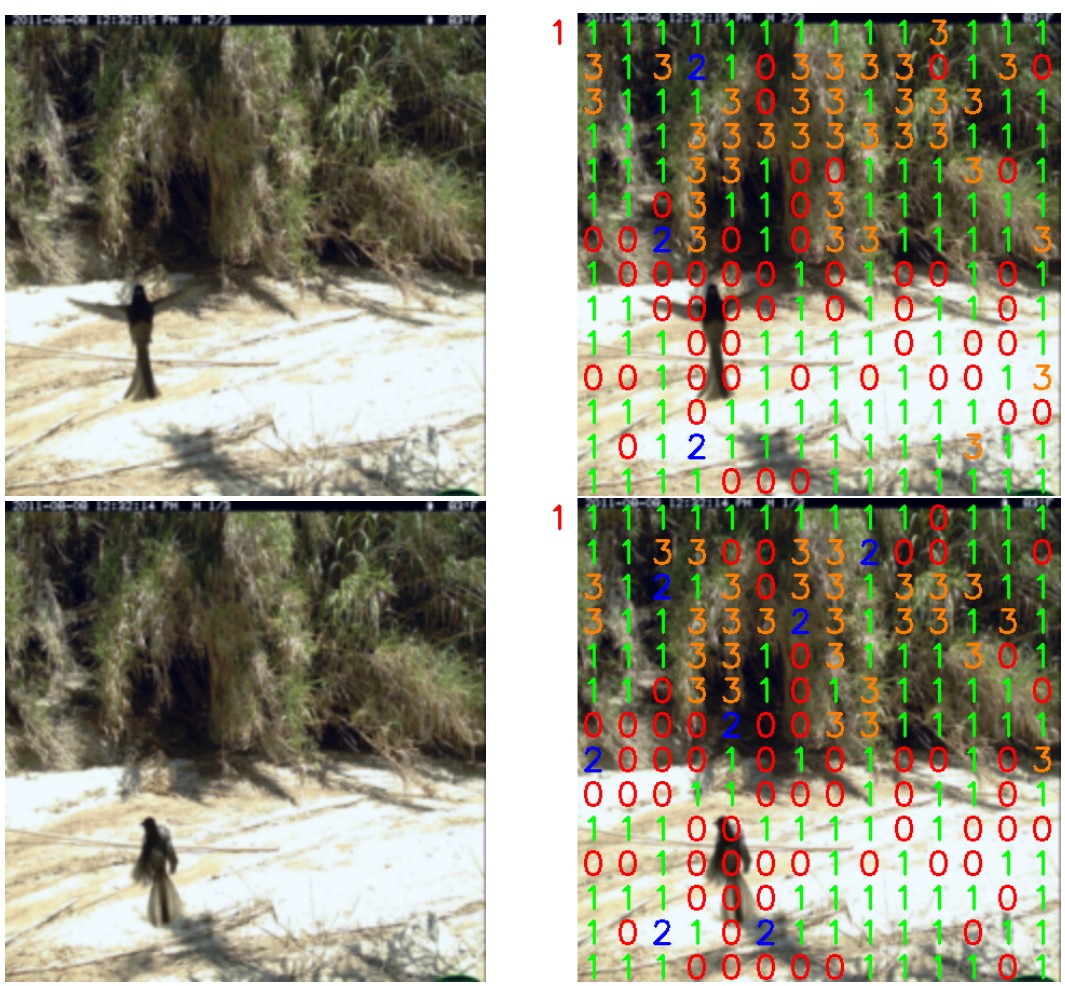

Location 43

Figure 10: Visualizations of routed indices of each patch in TerraIncognita (Beery et al., 2018) dataset. Left column is original image, and in right column we indicate where each patch is routed. Upper and lower images were took from same location but different time, therefore they have same background but different object (bird) shape and location.

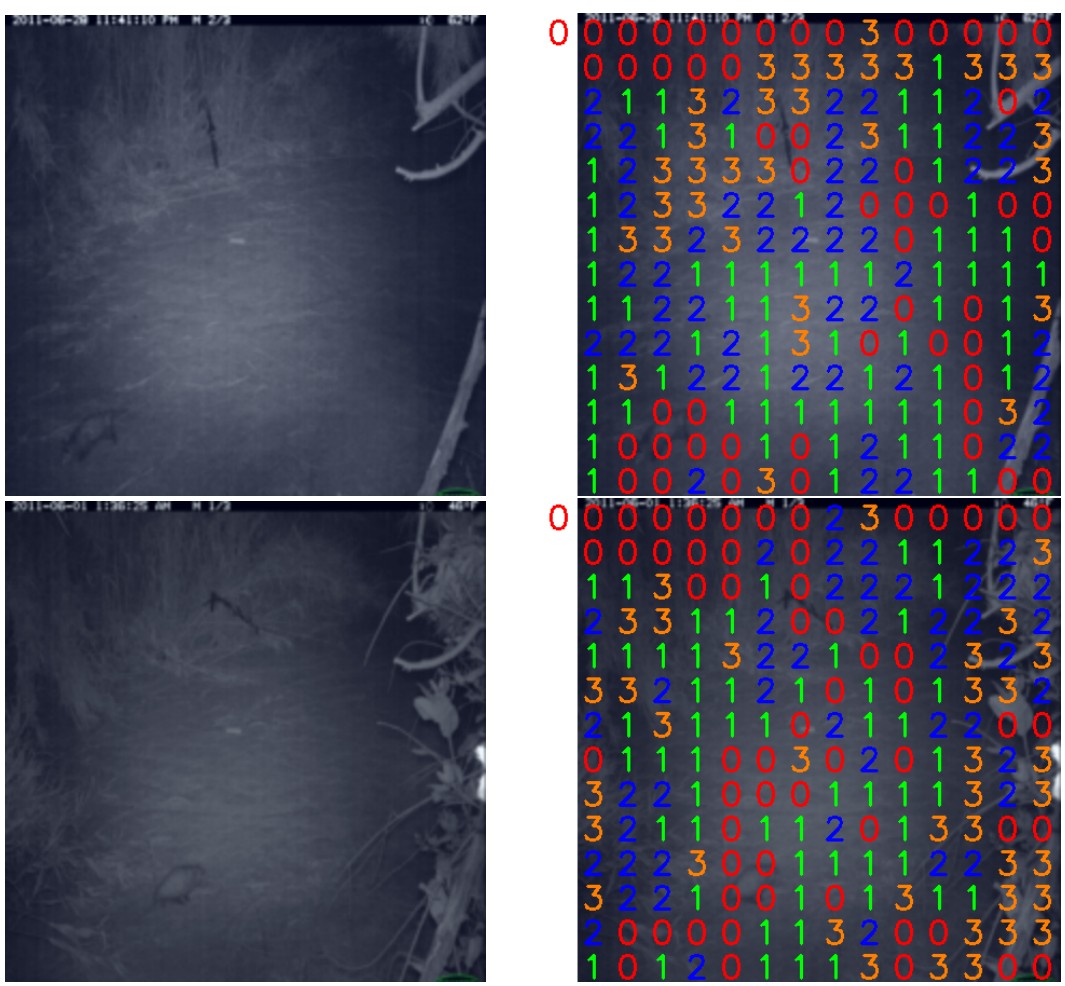

Location 46

Figure 11: Visualizations of routed indices of each patch in TerraIncognita (Beery et al., 2018) dataset. Left column is original image, and in right column we indicate where each patch is routed. Upper and lower images were took from same location but different time, therefore they have same background but different object (opossum) shape and location.

