# OpenReview forum: "Domain Generalization Using Large Pretrained Models With Mixture-of-Adapters"
_ICLR.cc/2024/Conference — ICLR 2024 Conference Withdrawn Submission_

### Official Review · Reviewer_6XsZ · 2023-10-27

**Soundness:** 2 fair
**Presentation:** 1 poor
**Contribution:** 1 poor
**Rating:** 3
**Confidence:** 4

**Summary:**

This paper investigates the performance of parameter-efficient finetuning methods (PEFT) and mixture-of-adapters (MoA) for the domain generalization task.  They argue that simple empirical risk minimization (ERM) on the source data without regularization performs well on the target domain data, providing a benefit compared to methods that require regularization or custom training schemes.  They argue that PEFT methods have a flatter loss landscape, and suggest this as an explanation for the generalization of such methods in this setting.

**Strengths:**

The strength of the method is that running the ERM algorithm with LoRA or KAdaption is easy to implement and doesn't require complicated hyperparameter tuning, and gives compelling performance out-of-the-box.  The analysis of the loss landscapes for PEFT methods in this context is a nice idea, and they provide some nice visualizations.

**Weaknesses:**

I have some qualms about the evaluation of the method in the paper.  First, there is no comparison with other works that study domain transfer using PEFT approaches, such as [1, 2, 3, 4] which use prompt tuning (a common PEFT approach).  The papers [1, 2, 3, 4] evaluate on the CORe50 domain transfer benchmark and [1, 4] evaluate on DomainNet (one of the datasets used in the paper under review).  Admittedly these papers are continual learning papers which is a slightly different setting, however since domain-incremental learning (seeing domains sequentially) is a harder task we can take the performance in these papers as a rough lower bound of what one should get in the setting of training on the union of all source domains "source combine", which is the setting studied by the authors in the present submission.  We note that when prompting the CLIP trained VIT-B encoder (note the submitted paper uses OpenCLIP which is very similar) the paper [4] achieves 67.0 accuracy on the unseen domains in DomainNet using the cumulative accuracy metric (which is a harder task than training on all domains then transferring to a single domain).  Meanwhile the present work gets around 62-63% on this same benchmark in an easier setting.  The earlier work [1] achieved 67.78 in the same evaluation setting when prompting both the vision and language encoder for CLIP (admittedly the prompt training of the language encoder comes at an additional training cost, but for inference the embeddings can be cached and thus this has a negligible effect on inference).

My second qualm is this paper compares against baselines that use a different pretraining e.g. SWAG IG3B.  The issue with this is the success of domain transfer depends critically on which representations you have from your backbone architecture.  If the backbone architecture has been trained on similar domains to the test data then the representations will transfer, however if there is a significant domain gap (e.g. evaluating on FGVC aircrafts using an ImageNet pretraining) then performance is poor.  In the case of domain transfer studied in this paper this issue becomes even more crucial, as there is not sufficient training data to cover the target domain.

[1] Yabin Wang, Zhiwu Huang, Xiaopeng Hong.  S-Prompts Learning with Pre-trained Transformers: An Occam's Razor for Domain Incremental Learning.  NeurIPS 2022.

[2] Benjamin Bowman et al. A-la-carte prompt tuning (APT): Combining
distinct data via composable prompting. CVPR 2023

[3] Zifeng Wang et al.  Learning to Prompt for Continual Learning. CVPR 2022.

[4] Julien Nicolas et al.  MoP-CLIP: A Mixture of Prompt-Tuned CLIP Models for Domain Incremental Learning.

**Questions:**

**Questions**

(Figure 3): You are plotting a spectral density using only 5 eigenvalues?  Why is the w/LoRA line flat?  What is the x-axis and the y-axis depicting?  There are no labels for either axis.

(Figure 2): Which two dimension of the loss landscape are you exploring in the figure?

Note: I like the patch visualizations, however I am not fully convinced of the argument of the routing corresponding to semantic information etc.  In my opinion it is hard to discover any visual patterns just by looking at the examples.  If there was a segmentation mask for the objects and a corresponding quantitative evaluation I would be convinced, however visually it is hard to make such an argument.

**Typos and Writing Issues**

“domain generalization (DG) algorithm aims” → “domain generalization (DG) algorithms aim”

“One of the most effective method” → “One of the most effective methods” (added s to methods)

“(a.k.a unseen domains)” → “(a.k.a. unseen domains)” (added period after second a in a.k.a.)

“to well predict on the target domain” → “to predict well on the target domain”

“shows that ERM algorithm” → “shows that the ERM algorithm”

“minimizes the loss on each domains” → “minimizes the loss on each domain”

“where K′ denotes the total number of training domains” → “where K’ is the number of target domains”

Add periods after paragraph titles “Effect of fine-tuning in pretrained models” and “Ablation study on routing strategy in Mixture-of-Adapter” for formatting consistency

---

### Official Review · Reviewer_JkEq · 2023-10-27

**Soundness:** 2 fair
**Presentation:** 1 poor
**Contribution:** 2 fair
**Rating:** 3
**Confidence:** 4

**Summary:**

This paper proposes a mixture-of-expert-based adapter fine-tuning method for the task of domain generalization. The idea is to adopt multiple adapters for a pretrained model and uses learnable routers to allocate each token to a proper adapter.

**Strengths:**

1. The finding that using different finetuning methods can lead to different performances is interesting.
2. The design of using routers to select an appropriate adapter for the token is reasonable.

**Weaknesses:**

1. The writing is poor. For example, just in the Intro. section, (1) the mixture usage of past tense (i.e. were, inspired...) and present tense is inappropriate; (2) the word Domain Generalization is abbreviated in the beginning but appears in its complete form many times; (3) there is no clear explanation for the last sentence in the first paragraph, the references here do not state the connection between DA and the necessity of exploring invariant features; (4) too many "and" in a sentence (i.e. we compare various aspects of different ...); (5) summarization of contributions in the Intro. section would be appreciated; (6) the citing format is also a mess in the paper. In all, I failed to grasp the motivation for the overall design and the connection between different sections after reading the manuscript. The authors are strongly suggested to doublecheck their manuscript before submitting it.

2. The statement in the intro. Section is questionable. The authors state that MSA+MLP outperforms MSA in Terraincognita due to a large distribution shift. However, the definition of this "large distribution shift" is not clear enough. This dataset contains images taken from different views, according to [a], the shift is similar to PACS. Maybe there are other reasons that MSA+MLP can outperform MSA in this dataset.

3. The overall method in Sec. 5 seems to be a combination of existing ideas. To highlight their contributions, the authors may present their methods with more details and include comparisons with others.

[a] OoD-Bench: Quantifying and Understanding Two Dimensions of Out-of-Distribution Generalization, in CVPR'22

**Questions:**

See weakness

---

### Official Review · Reviewer_PB7i · 2023-10-30

**Soundness:** 1 poor
**Presentation:** 1 poor
**Contribution:** 1 poor
**Rating:** 1
**Confidence:** 4

**Summary:**

This work analyses use of PEFT (parameter efficient finetuning) methods to the domain generalization setting and explores them together in a mixture-of-experts setup.

In particular: it points out that the PEFT method used can have an impact on DG. It analyses loss landscape and points that PEFT methods reach a more generalizable optimization point. Show results that KAdaption performs the best of other methods. Introduce MoA (mixture of adapters).

The results are based on 5 vision tasks.

**Strengths:**

The initial observation that PEFT methods have a great impact on domain generalisation seem valid and worth exploring, however the rest of the paper does not follow or read clearly.

**Weaknesses:**

There are many weaknesses in this work, the main ones being the flatness results and weak MoA results. Following their presence in the work:

A) “Picking the proper regularisation depending on distribution shift”, both figure1 and table 1 show a story of picking Bias (MSA+MLP) is the best (or close to) in the 5 settings and that KAdaption beats LoRA,r=2 in all settings. This is very different from one having to carefully pick the PEFT setting depending on the task distribution shift (which is what I read from the text).

B) Analysis of loss landscape flatness: to the best of my understanding this whole section is completely flawed. The root problem here is that the flatness and eigen values across such different network parameterizations can’t be compared. As [1] points out: “if we allow to reparametrize a function, the geometry of its parameters can change drastically without affecting its generalization properties”.

C) MoA results seem to indicate that in 4 out 5 datasets there is no difference between KAdaption-MoA and KAdaption. In the OfficeHome dataset that difference is: 90.3->90.6. In my opinion this is weak evidence to justify MoA.

[1] Dinh, Laurent, et al. "Sharp minima can generalize for deep nets." ICML 2017

**Questions:**

See weaknesses.

---

### Official Review · Reviewer_Ksvd · 2023-11-02

**Soundness:** 2 fair
**Presentation:** 3 good
**Contribution:** 2 fair
**Rating:** 5
**Confidence:** 4

**Summary:**

This paper finds that the use of adapters in parameter-efficient finetuning (PEFT) methods serves as an effective regularization for domain generalization (DG) tasks, and existing adapter methods for large models such as LoRA and KAdaptation achieve superior performance on DG benchmarks. To choose the optimal amount of regularization, this paper proposes a mixture-of-expert-based adapter finetuning method (MoA), which employs multiple adapters that have varying capacities and allocate each token to proper adapters by learnable routers. Using both PEFT and MoA, the proposed method outperforms some state-of-the-art methods on diverse DG benchmarks.

**Strengths:**

* The finding that PEFT methods with adapters can help improve the domain generalization ability of pre-trained models may give some insights for future studies.

* The idea of choosing the optimal amount of regularization due to the strength of distribution shifts makes some sense, and the design of the mixture-of-adapters is also interesting in some way. (However, there are also some concerns regarding this, which will be mentioned in the Weakness part.)

* The proposed method is evaluated on several DG benchmarks and outperforms some SOTA methods and models.

**Weaknesses:**

* There are some concerns regarding the design of MoA. (1) The motivation for using different amounts of regularization comes from the experiments with different bias tuning. However, the main experiments use other adapter-based methods such as LoRA and KAdaptation. It is not sure whether these methods would have the same finding as bias tuning. There is also no experiment to evaluate the performance of adapters with different capacities. (2) From the results in Table 2, the improvement of MoA seems to be minor compared with using one single adapter.

* The phenomenon that fully finetuning may cause overfitting and harm DG abilities has already been studied in some previous works, such as (Gao et al., 2021, Kumar et al.,2022) mentioned in the paper. So, the result that PEFT can help mitigate this problem is not very surprising since it is designed to control the changes from the pre-trained model. Besides, the PEFT and the mixture-of-experts methods in the paper are all from existing works. Considering this, the overall contribution of this paper is a little insufficient.

* The PEFT and MoA results are based on the CLIP pre-trained model, while other SOTA results mentioned in the paper are based on the SWAG model, which may make the comparison unfair. Besides, this paper only evaluates the CLIP pre-trained model, and the performance of PEFT and MoA on other types of pre-trained models is unknown.

* There are some other methods to mitigate overfitting during finetuning, which may be related to this work, such as [1] and [2]. These related methods may need to be discussed and compared in the experiments.

[1] Finetuning can distort pretrained features and underperform out-of-distribution

[2] Explicit Inductive Bias for Transfer Learning with Convolutional Networks

**Questions:**

Why should MoA use token-wise routing? Figure 5 shows that similar tokens are routed to the same expert, but how is it related to the motivation of distribution shifts?